# Role of the Outer Metal of Double Metal Cyanides on the Catalytic Efficiency in Styrene Oxidation

**Paulina Molina-Maldonado [1], Rosario Ruíz-Guerrero [2,*] and Carlos Hernández-Fuentes [2]**

[1]   Instituto Politécnico Nacional, Centro de Investigación en Ciencia Aplicada y Tecnología Avanzada-Unidad Legaria, 11500 CDMX, Mexico; mompau@gmail.com
[2]   Instituto Politécnico Nacional, Centro de Investigación e Innovación Tecnológica, 38010 CDMX, Mexico; carlosfhef@gmail.com
*   Correspondence: rosarior@hotmail.com; Tel.: +52-5329-6000 (ext. 68325)

**Abstract:** The catalytic efficiency of double metal cyanide (DMC) has been shown to be very effective in heterogeneous catalysis. The catalytic activity of the outer divalent cations (Mn, Co, Ni, and Cu) of a family of hexacyanocobaltates was examined in the oxidation reaction of styrene, as a model molecule, using tert Butyl Hydroperoxide (TBHP, Luperox) as an oxidizing agent. The most electronegative outer cations showed the best conversions, with 95% for copper, followed by nickel with 85% conversion of the monomer at atmospheric pressure and temperature of 75 °C. The evidence showed that the catalytic activity and selectivity towards oxidized products are strongly linked to the accurate choice of the outer cation in the DMC together with the oxidizing agent.

**Keywords:** hexacyanocobaltates; oxidation process; TBHP; transition metal; reaction mechanism; synthesis of organic molecules

## 1. Introduction

Double metal cyanides (DMCs) are coordination complexes formed from the assembly of two transition metals by cyanometal structures $[M'(CN)_n]$. This means that an internal cation (M') remains linked to the CN ligand at the carbon end, while the outer metal forms a coordination bond at the N end. Hence, its molecular formula can be described as follows: $M_u[M'(CN)_n]_v \cdot xH_2O$. These materials have received great attention for their variety of interesting properties, such as photomagnetism [1,2] magnetic pole inversion [3,4], their possible technological applications in hydrogen storage [5], battery materials [6], and electro-photocatalysis [7] or catalysis [8–10]. Nowadays, the wide spectrum of applications that these materials can have, directs their syntheses to structures with well-defined properties for a specific use. The hexacyanocobaltates are materials belonging to the family of DMCs with Co as the internal cation. Their structures are based upon the cubic $M^{II}_3[Co^{III}(CN)_6]_2 \cdot xH_2O$ framework illustrated in Figure 1, assembling the molecular block $[Co(CN)_6]^{3-}$ through the metal ($M^{II}$), which links neighbouring blocks at its N ends. In the hexacyanocobaltates, the charge balance with the $M^{II}$ ions leads to vacancies close to 33% of the $[Co(CN)_6]^{3-}$ sites in the framework [11]. Consequently, the water molecules complete the octahedral coordination sphere of the $M^{II}$ ions.



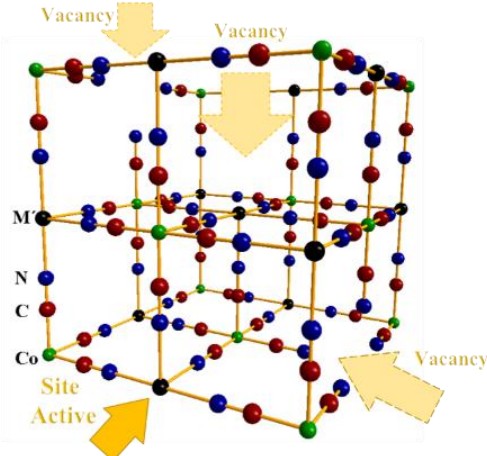

**Figure 1.** Unit cell for the structure of hexacyanocobaltates $M_3[Co^{III}(CN)_6]_2$. where M: Mn, Co, Ni and Cu (black), $Co^{III}$ (green), C (red), and N (blue) atoms, respectively. The vacancies have been represented by the absence of the octahedral block, and the water solvate molecules have been omitted for clarity. (For interpretation of the references to color in this figure legend, the reader referred to the web version of the article.)

Interest in hexacyanocobaltates has focused on their catalytic applications [12–14]. The Co(III) ion, in its low-spin state, has six electrons in the $t_{2g}$ orbitals, and these orbitals are filled. This provides high stability to the block $[Co(CN)_6]$, and no obvious participation of the hexacyanocobaltate (III) anion in oxidation or reduction processes are expected. On the other hand, when it comes to bridges with a Co–C≡N–M type chain, a second cation (M) is attached to the CN unit, and the bond to N atoms of the cyanide bridge is key. This orientation is particularly important for deducing the spin state of the outer cation (M), while the C end is a strong-field ligand and, therefore, produces a low-spin ion. The N end is a weak-field ligand and, thus, stabilizes the high-spin ion states. In this way, is possible the participation of this second cation incorporated as an outer cation in the catalytic processes.

The reported catalytic activity in this family of compounds shows a certain dependence on the involved metal (M) [15], an effect that deserves to be studied since it could be used as a way of tuning the catalytic properties of these materials. The outer cations of this type of compound maintain mixed valences [16,17], which can act as mediators in the transfer of electrons [18]. Likewise, all M metals are sited at the pore surface, with the possibility of direct interaction with the reactants molecules in the reaction medium. This could be used as a way of sensing the catalytic activity of the catalyst.

In this contribution, the properties of divalent transition-metal hexacyanocobaltate (III) (M = $Mn^{2+}$, $Co^{2+}$, $Ni^{2+}$, and $Cu^{2+}$) as a catalytic material are evaluated, correlating the outer cation with the oxidation of styrene, with the help of the soft oxidizing agent, tert Butyl Hydroperoxide (TBHP, Luperox). This reaction is of considerable importance because styrene is an essential intermediate in the production of a number of fine chemicals and pharmaceuticals.

The obtained results shed light on the nature of the role of the metal in the catalytic properties of the material. To the best of our knowledge, this is the first study on this subject for this family of materials.

As for the catalytic oxidation reactions in olefins promoted by hexacianometalates (III) of divalent transition metals the studies available are scarce. Although it has been reported that there are a probable intermediary species that function as a transporter of the oxygen atom to the alkene [19], the role of the outer cation has not been determined.

## 2. Results and Discussion

### 2.1. Catalyst Characterization

#### 2.1.1. Infrared (IR) Spectroscopy

The nature of the obtained hexacyanocobaltate (III) catalyst was corroborated from infrared spectra [20]. Figure 2 shows a shift of the cyanide band frequency $v$(CN) for the family of catalysts studied between 2128 cm$^{-1}$ for K–Co and 2187 cm$^{-1}$ for Cu–Co (Table 1). The variations in the frequency of the C≡N bond-stretching band for the Co–C≡N–M bridge are associated with the nature of the cations.

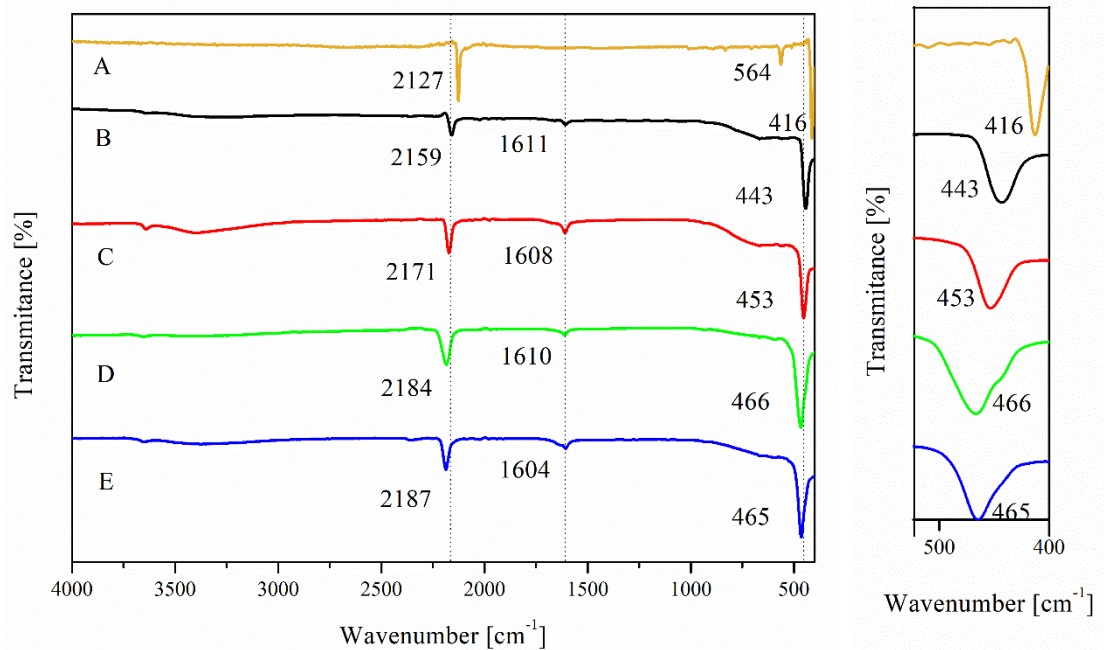

**Figure 2.** Infrared (IR) spectra of (A) K$_3$[Co(CN)$_6$], (B) Mn–Co, (C) Co–Co, (D) Ni–Co, (E) Cu–Co catalysts.

**Table 1.** Overview of synthesis identification.

| Catalyst [a] | Identification M–Co | $v$ (CN) cm$^{-1}$ | $\delta$(OH) [c] cm$^{-1}$ | $v$ (Co−CN) cm$^{-1}$ | $\delta$(Co−CN) cm$^{-1}$ |
|---|---|---|---|---|---|
| K$_3$[Co(CN)$_6$]* [b] | K–Co | 2127 | - | 564 | 416 |
| Mn$_3$[Co(CN)$_6$]$_2$·11H$_2$O | Mn–Co | 2159 | 1611 | 443 | - |
| Co$_3$[Co(CN)$_6$]$_2$·13H$_2$O | Co–Co | 2171 | 1608 | 453 | - |
| Ni$_3$[Co(CN)$_6$]$_2$·H$_2$O | Ni–Co | 2184 | 1610 | 466 | - |
| Cu$_3$[Co(CN)$_6$]$_2$·9H$_2$O | Cu–Co | 2187 | 1604 | 465 | - |

[a] The reported hydration degree was stimmed from TG curves; [b] Precursor material evaluated as a blank; [c] bending vibration of coordinated water molecules.

In these materials, the cyanide ligand donates its sigma (nonbonding) electrons to the metal, while accepting electron density from the metal through the overlap of a metal t$_{2g}$ orbital and a ligand $\pi$* orbital. Overlapping produces a lower energy molecular $\pi$ orbital, also known as $\pi$-back bonding, which is responsible for the negative charge transfer of the binder block-*s* to the metal $\pi$. The ligand is thus acting as a $\sigma$-donor and a $\pi$-acceptor (Figure 3).

The stability of the block with the inner metal [Co(CN)$_6$] is greater than that obtained by the outer cation within the M–CN–M′ chain, but since the Co is always the internal metal, the outer one will be responsible for the modification in physicochemical behaviour.

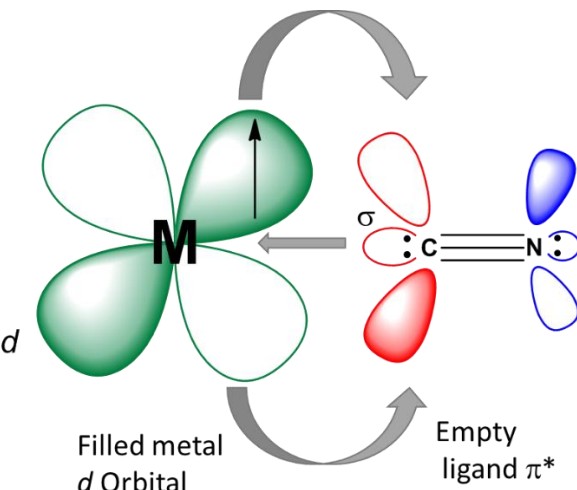

**Figure 3.** Representation of π back-bonding responsible for the transfer of negative charge from the binder block to an orbital π.

Greater π retro-donation is evident in the infrared (IR) spectra, when π-bonding of M → CN increases, the strength of the C≡N bond decreases and, therefore, ν(CN) decreases. Likewise, the increase in ν(CN) observed in the different catalysts (see Table 1) in the sense Cu–Co > Ni–Co > Co–Co > Mn–Co obey generally; the decreasing number of *d* electrons of the outer cation and can be attributed to a decrease in metal-cyanide π-bonding. In addition, the nitrogen lone pair of CN⁻ resides in mostly a C–N antibonding orbital; therefore, the involvement of the nitrogen lone pair of electrons in bonding increases the strength of the carbon–nitrogen bond, which results in an increase in ν(CN) for bridging cyanide.

An additional M–CN stretching vibration can be observed in the IR spectra, with a shift towards higher frequency values in the band close to 400 cm$^{-1}$ (Figure 2). The frequency of the δ(M–CN) band it grows; 443 cm$^{-1}$ for Mn–Co, 453 cm$^{-1}$ for Co–Co, 465 cm$^{-1}$ for Cu–Co, and 466 cm$^{-1}$ for Ni–Co in the sense of electronegativity of outer cation [21], The relative electronegativity present in the outer cation provides a Lewis acid character that is characteristic of the cation. Likewise, the stability of a metal complex, DMC in this case (strength of the metal-ligand bond) should be the function of the electron-attracting power of the metal. Consequently, we may identify the electronegativity´s for divalent metal ions by the Irving- Williams order [22], according to which the stability of complexes increases in the order: $Mn^{2+} < Co^{2+} < Ni^{2+} < Cu^{2+}$.

This trend in electronegativity's, in the same order, can be an ally to interpret the affinities of these metal ions present in DMC as outer cations to attract the oxidizing agent (TBHP) and thus promote a catalytic cycle.

2.1.2. Energy-Dispersed X-ray Spectrometry (EDS) and Hydration Degree

The atomic metal ratio was estimated by Energy-Dispersed Spectroscopy (EDS) analyses (Figure 4). The M:Co ratio in all catalysts, was close to 3:2, in correspondence with the expected formula of $M_3[Co(CN)_6]_2 \cdot xH_2O$, except for the sample where M is also Co (Figure 4b), here the Co is the only cation detected.

The low potassium content indicates the presence of traces of this ion originally present in the precursor salt $K_3[Co(CN)_6]$. Excess carbon in all samples is expected according to the use of graphite tape in the measurement method.

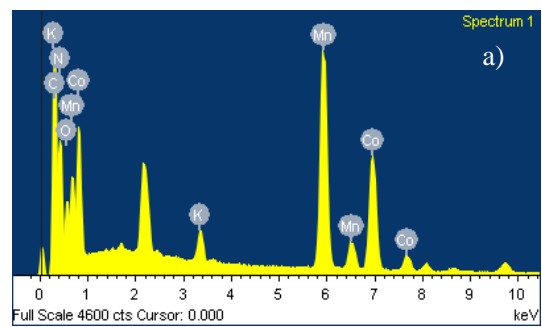

| Element | Weight% | Atomic% |
|---------|---------|---------|
| $C_K$ | 29.77 | 41.45 |
| $N_K$ | 36.17 | 43.19 |
| $O_K$ | 6.93 | 7.25 |
| $K_K$ | 0.78 | 0.33 |
| $Mn_K$ | 14.59 | 4.44 |
| $Co_K$ | 11.75 | 3.33 |
| Totals | 100.00 | 100.00 |

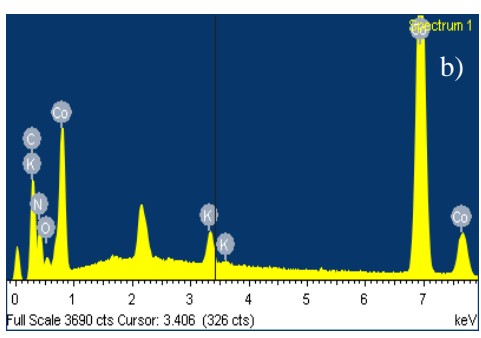

| Element | Weight% | Atomic% |
|---------|---------|---------|
| $C_K$ | 28.86 | 49.67 |
| $N_K$ | 20.54 | 30.31 |
| $O_K$ | 2.17 | 2.80 |
| $K_K$ | 1.25 | 0.66 |
| $Co_K$ | 47.18 | 16.55 |
| Totals | 100.00 | 100.00 |

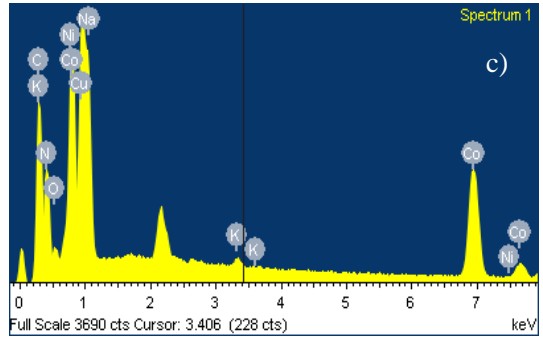

| Element | Weight% | Atomic% |
|---------|---------|---------|
| $C_K$ | 32.04 | 43.2 |
| $N_K$ | 38.52 | 44.55 |
| $O_K$ | 3.13 | 3.16 |
| $K_K$ | 4.96 | 3.4 |
| $Co_K$ | 11.80 | 3.24 |
| $Ni_K$ | 9.55 | 2.44 |
| Totals | 100.00 | 100.00 |

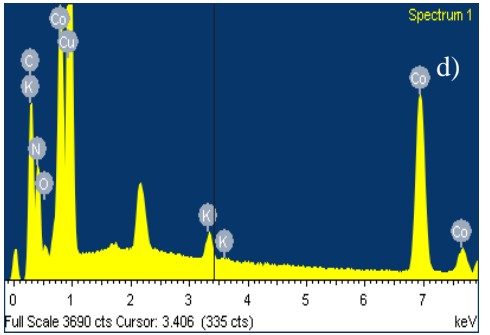

| Element | Weight% | Atomic% |
|---------|---------|---------|
| $C_K$ | 30.39 | 46.67 |
| $N_K$ | 30.60 | 40.29 |
| $O_K$ | 1.47 | 1.70 |
| $K_K$ | 0.64 | 0.30 |
| $Co_K$ | 14.51 | 4.54 |
| $Cu_K$ | 22.39 | 6.50 |
| Totals | 100.00 | 100.00 |

**Figure 4.** Images of (**a**) Mn–Co, (**b**) Co–Co, (**c**) Ni–Co, and (**d**) Cu–Co samples. Peak at 2.15 keV corresponds to Au, used in pre-treatment of measurement.

An analysis of the curves in Figure 5 shows that the dehydration temperature in the materials analyzed depends on the outer cation. This has of course to do with the different polarizing capabilities of each cation and its ionic radius. The moisture that is lost corresponds to the water molecules coordinated to the metal bound to the N terminal. The water molecules present are evacuated from the structure at a temperature below 100 °C. This indicates that zeolitic and coordinated waters are desorbed in a single stage.

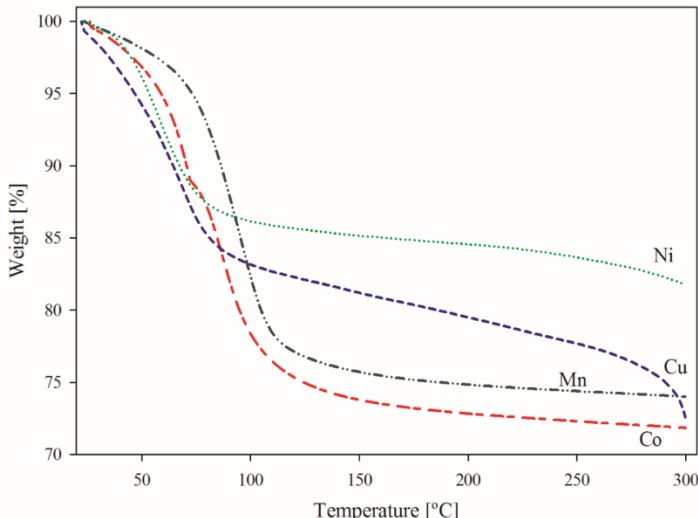

**Figure 5.** Thermal Gravimetric Analysis (TGA) of Mn(black), Co (red), Ni (green), and Cu (blue).

The lowest temperature values of dehydration correspond to Cu-Co followed by Ni-Co, Co-Co and finally Mn-Co. This suggests that copper is the outer metallic atom with the lowest positive charge density within the studied followed by the others in the same order.

At lower positive charge in the metal cation the polarizing power decreases ($Ze/r^2$) and the ability to retain water molecules in its environment [23].

### 2.1.3. X-ray Diffraction (XRD) and Scanning Electron Microscopy (SEM) Images

Figure 6 shows the powder X-ray diffraction (XRD) patterns of Mn–Co, Co–Co, Ni–Co and Cu–Co catalysts. Their crystal structures were confirmed. We observed very high crystallinity in the materials and confirmed their cubic-lattice structure, typical of a double metal cyanide, according to reported crystal structures for the hexacyanocobaltates (III) in XRD databases, such as ICSD 00-051-1898, ICSD 01-071-0807, ICSD 01-089-3738, and ICSD 00-051-1895.

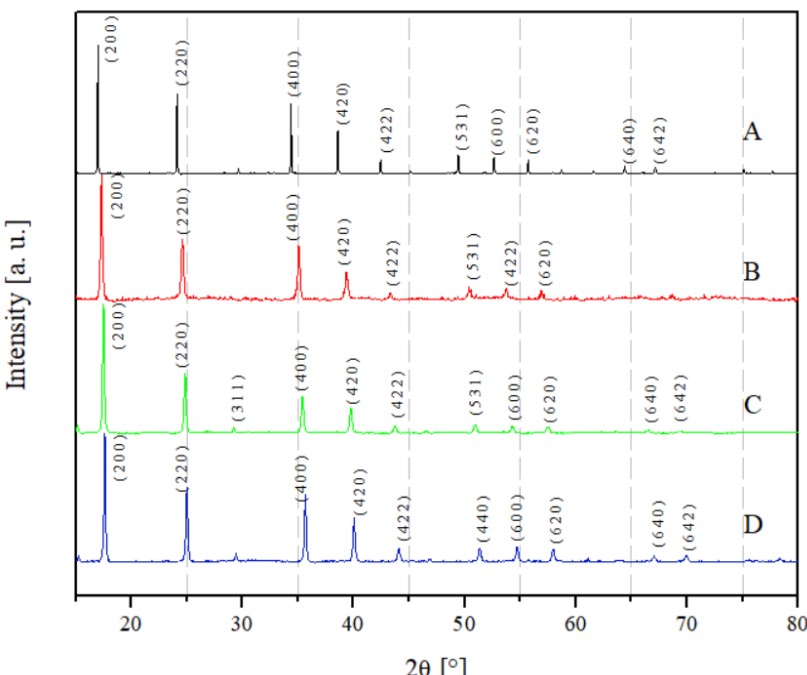

**Figure 6.** X-ray diffraction (XRD) spectra of (A) Mn–Co, (B) Co–Co, (C) Ni–Co, and (D) Cu–Co catalysts.

The patterns for Mn–Co, Co–Co, and Cu–Co take to the Fm-3m space group, while Ni–Co shows a space group of F-43m, all with octahedral coordination. When the outer metal adopts this type of coordination, a highly symmetric structure based on a cubic unit cell is obtained, where 1/3 of the molecular block sites remain vacant [24]. The vacancies obtained have pore sizes defined in small windows that can be interconnected.

Additionally, the morphology and structure of the prepared Mn–Co, Co–Co, Cu–Co, and Ni–Co samples were further investigated using Scanning Electron Microscopy (SEM). Figure 7 shows the SEM images of each of these samples. Figure 7a shows the SEM image of optimized Mn–Co. It can be seen that these $Mn_3[Co(CN)_6]_2$ crystals have uniform cubic shapes that correspond to single crystals of $Mn_3[Co(CN)_6]_2$ with cubic structures. The distances between the lattice planes can be determined, and the results are in agreement with the XRD data. The sizes of these cubic shapes are in the range of 2.0–2.5 μm, the largest within the group of catalysts synthesized. Figure 7b shows the SEM images of $Co_3[Co(CN)_6]_2$ nanostructures prepared with the same conditions as $Mn_3[Co(CN)_6]_2$, which did not show the same size, morphology, or crystallinity. Figure 7c shows the SEM image of $Ni_3[Co(CN)_6]_2$, and Figure 7d shows the SEM image of $Cu_3[Co(CN)_6]_2$ nanocrystals.

As part of the contribution of the study of these materials, it has already been reported before the pore volume (Vp) [25] that present the following order in terms of size; Mn–Co > Co–Co > Cu–Co > Ni–Co (Table 2). One of the main characteristics of these materials is the small pore windows that are formed from the absence of the blocks. It is worth noting that the role of the vacancies is important because they suppress the perfectly cubic structures and increase the surface of the catalysts. In this way, both the crystallite size and Vp could be indicators of the catalytic performance of the different materials.

**Table 2.** Crystallite size according to the Scherrer equation and *Vp*, the pore volume estimate from $N_2$ and $CO_2$ isotherms.

| Catalyst | Scale | Crystallite Size Å [a] | Adsorbate | *Vp* [25] (cm³/g) |
|---|---|---|---|---|
| Mn–Co | Bulk | 1087 ± 0.50453 | $N_2$ | 0.448 ± 0.012 |
| | | | $CO_2$ | 0.281 ± 0.012 |
| Co–Co | Nanometer | 414 ± 0.0542 | $N_2$ | 0.381 ± 0.002 |
| | | | $CO_2$ | 0.224 ± 0.002 |
| Ni–Co | Nanometer | 632 ± 0.02268 | $N_2$ | - |
| | | | $CO_2$ | 0.181 ± 0.003 |
| Cu–Co | Nanometer | 392 ± 0.03907 | $N_2$ | 0.234 ± 0.003 |
| | | | $CO_2$ | 0.202 ± 0.001 |

[a] Calculation made with 5 diffractions maxima of greater intensity in each sample.

Cu–Co presented a small crystallite size and small Vp with respect to its Co–Co and Mn–Co homologs. The vacancies in this catalyst allow for adequate diffusion of $N_2$ during adsorption–desorption studies, with which it is possible to estimate Vp. This same estimate using $CO_2$ as an adsorbate decayed slightly. It should be noted that one of the main characteristics in this material is its strong polarizing power due to the outer copper cation [25], which is responsible for an important interaction with other polarizable host molecules with which it could interact, in our case a better interaction whit the olefin.

Ni–Co presented the smallest crystallite size of the group and, as a result, its Vp is also the smallest. This hinders the accessibility of $N_2$ during adsorption–desorption studies, causing very low diffusion and making the estimation of Vp impossible. However, estimation of Vp in the presence of $CO_2$ is viable, which allows us to verify the small volume with respect to its homologs of the family. The feasibility of adsorption presented by Ni–Co against $CO_2$ is probably due to the interaction of the outer cation with the oxygen atom of $CO_2$. This is possibly due to the particularity of the $CO_2$ molecule, which presents an elongated form and, at both ends, contains an oxygen atom whose electronic pair maintains an interaction with Ni, either by the proximity with which they are and/or by the high disposition of these.

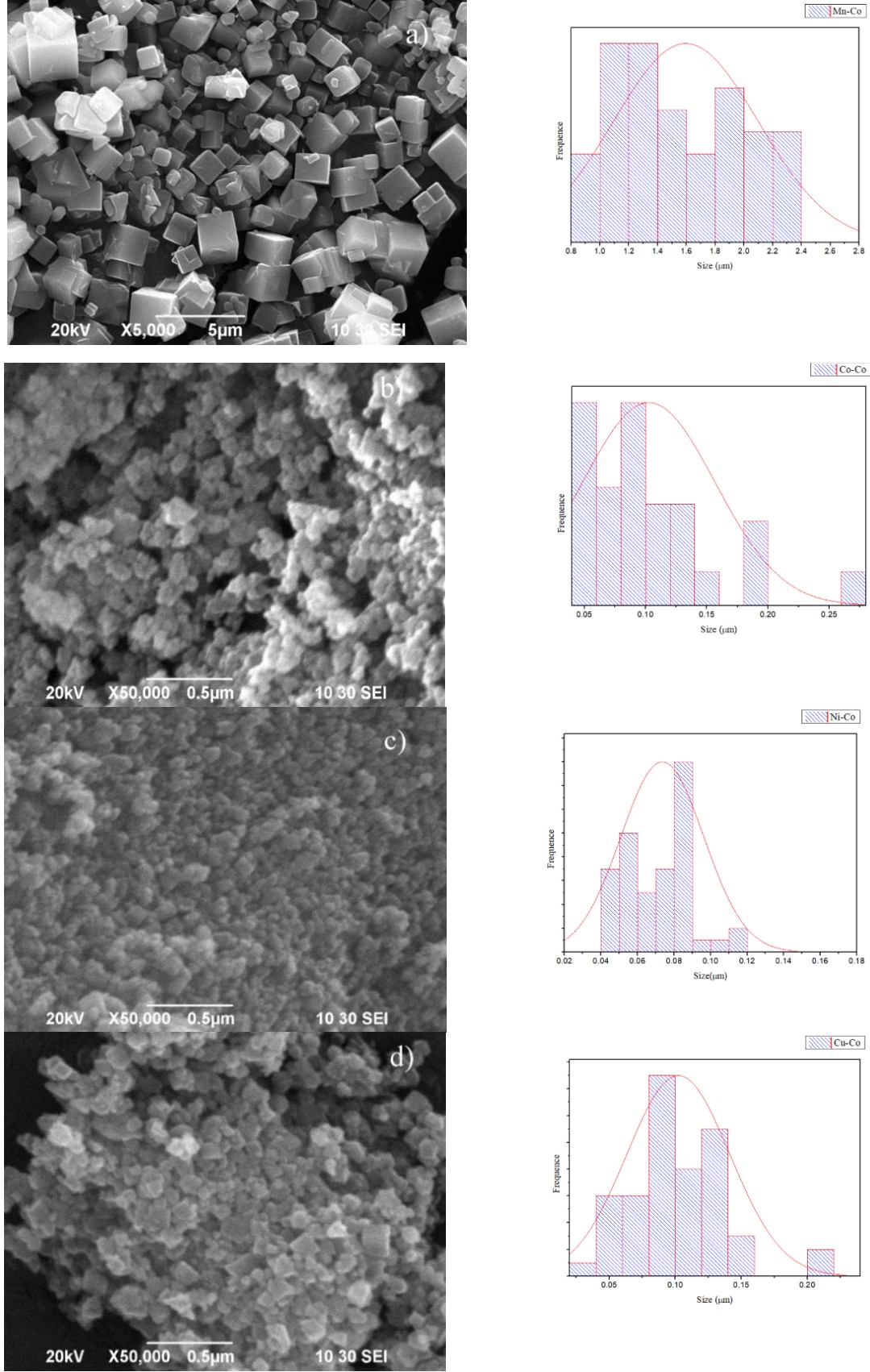

**Figure 7.** Scanning Electron Microscopy (SEM) images of (**a**) Mn–Co, (**b**) Co–Co, (**c**) Ni–Co, and (**d**) Cu–Co samples.

In the case of Co–Co, both the size of the crystallite and Vp allow for free access to the windows formed in the structure of any of the adsorbates ($N_2$ or $CO_2$). However, it seems that the outlet cation (Co) for this material only works as a load stabilizer. Finally, in the case of Mn–Co, Vp is so large (compared to the other materials) that it allows for estimation of properties with both gases. These results are, however, considerably different.

It is known that the catalytic performances of these materials are related to structural and chemical factors. Additionally, in heterogeneous catalysis the size and shape-dependent factors are closely related to the surface chemistry of the catalyst, which essentially determines the catalytic performance. In this sense, as the hexacyanocobaltates have well defined cubic structures, in the present study the outer cation is changed, adding own characteristics to each material and it will be this cation that must differentiate the catalytic performance predominantly.

Co–Co, Ni–Co, and Cu–Co have crystallite sizes together to Vp values relatively proportional among them, so it is not justified that these parameters are a determining factor in their catalytic performance. In this sense, we consider that it is the nature of the outer cation in direct interaction with an oxidizing agent, such as TBHP (highly polarizable), that dictates its performance.

## 2.2. Catalytic Performance of DMC in the Styrene Oxidation and Proposed Mechanism

The catalytic activities of these samples and the materials used as references (blank tests) were tested in the oxidation of styrene with TBHP. Acetonitrile was the solvent of choice, as it has shown optimal performance in selective oxidations [26,27]. Besides the conversion of styrene and TBHP, the Table 3 shows the selectivity of styrene towards the different oxidized products.

**Table 3.** Performance of hexacyanocobaltates complexes $M_3[Co^{III}(CN)_6]_2$ on styrene by tert Butyl Hydroperoxide (TBHP).

| Entry | Catalyst | Reaction Time (h) | Conversion (%) | | Selectivity | | | Yield of SO (%) |
| | | | Styrene | TBHP | SO | Bzh | Acid [c] | |
|---|---|---|---|---|---|---|---|---|
| 1 | Mn-Co [a] | 4 | 30.8 | 30.5 | 2.2 | 32.1 | 65.7 | 0.6 |
| 2 | Mn-Co [a] | 10 | 35.7 | 35.6 | 2.8 | 39.3 | 57.9 | 0.8 |
| 3 | Co-Co [a] | 10 | 35.6 | 35.3 | 2.5 | 33.9 | 63.6 | 0.8 |
| 4 | Ni-Co [a] | 10 | 42.3 | 51.2 | 10.2 | 49.9 | 39.9 | 4.2 |
| 5 | Cu-Co [a] | 10 | 43 | 71 | 36.3 | 47.9 | 15.8 | 20.5 |
| 6 | Mn-Co [b] | 10 | 45.3 | 30.5 | 26.6 | 73.4 | - | 12 |
| 7 | Co-Co [b] | 10 | 43 | 28.9 | 24.9 | 75.1 | - | 10.7 |
| 8 | Ni-Co [b] | 10 | 88.6 | 61 | 52.4 | 47.6 | - | 46.4 |
| 9 | Cu-Co [b] | 10 | 95.1 | 89.6 | 67.4 | 32.6 | - | 64 |
| 10 | $K_3[Co(CN)_6]$ [b] | 10 | 44.1 | 32 | 21.1 | 78.8 | >1 | 10 |
| 11 | $Cu(NO_2)_6 \cdot 6H_2O$ [b] | 4 | 68.3 | 100 | 30.3 | 58.7 | 11 | 21.5 |
| 12 | Blank | 8 | 34 | 23.5 | - | 77.1 | 22.9 | 0.0 |

Reaction conditions: Catalyst/styrene/TBPH/$CH_3CN$: [a] 0.008 mmol/1 mmol/1.5 mmol/5 mL, [b] 0.08 mmol/1 mmol/ 1.5 mmol/5 mL. Reaction temperature: 75 °C. [c] Benzoic acid.

Initially, the group of four catalysts with different outer cations (Mn, Co, Ni and Cu) were evaluated in catalytic reactions with low catalyst concentrations: 0.008 mmol of catalyst by 1.0 mmol of styrene (I), (entries 1–5 of Table 3). In all cases, two products were obtained mostly, besides the primary product: *i*) benzaldehyde (II), with one carbon atom less than the starting reactant, *ii*) benzoic acid (III); which is formed from the consecutive oxidation of the aldehyde [28] (Scheme 1). The reaction of Entry 2 confirms how II is transformed into III as the hours pass, while the oxidizing agent is active.

These reactions (Entry 1–5) advanced slowly with moderate substrate conversions, 30 to 40% and with significant selectivity's towards II, without seemingly passing through the epoxide (IV), which is normally assumed as its precursor [28]. The formation of III in these reactions was monitored, showing selectivity's above 50% in most cases, indicating that the oxidizing agent still remains active.

**Scheme 1.** Oxidation of styrene to benzoic acid in the presence of TBHP (tert-butyl hydroperoxide) whit low % cat.

Next, the effects of the amount of catalyst were examined. The optimized value for obtaining styrene oxide (IV) with one of the catalysts (Cu-Co) resulted in 8% of catalyst relative to the initial styrene: 0.08 mmol of catalyst, 1.0 mmol of styrene (Thus, entries 6–9 corresponded to the evaluation of the different catalysts with these conditions.

The results for this group of reactions also show the exclusive formation of two products, but surprisingly this time: styrene oxide (IV) and Benzaldehyde (II), the one probably was formed from direct oxidative cleavage of styrene by TBHP, without participation of the catalyst, as in Entry 10.

The presence of IV in these reactions (Entries 6–9), reveals the unimportant role that the catalyst could play in the first group of reactions (Entries 1–5) due to their low concentration and therefore low participation to form the epoxide with important yields. Entry 12 strengthens these hypotheses and confirms the smooth oxidation of olefin to benzaldehyde, in the absence of a catalyst with its subsequent oxidation to carboxylic acid, see Scheme 2. It should be mentioned that TBHP has demonstrated the ability to directly oxidize C-H bonds in the absence of catalyst [29].

**Scheme 2.** Transformation of styrene in the presence of TBHP whit 0.08 mmol cat.

The participation of the catalyst is evident in the preferential oxidation towards IV. Cu and Ni are the cations who maintain a predominant oxidation to oxirane. The effect that has a higher concentration of the catalyst in the reaction evidences its participation in the catalytic cycle that revels the possible presence of a catalytic intermediate-oxidizing agent (V) that is favored and that limits the consecutive oxidation of II to III. Scheme 3 describes the mechanism proposed for this purpose.

**Scheme 3.** Possible mechanism for the metal-hexacyanocobaltate with TBHP. The red spheres are inner Co bonded to C atoms, and the grey atoms are the outer metals bonded to N atoms.

The mechanism proposed in Scheme 3 is based on the results obtained. The oxidizing agent (TBHP) reacts with the active surface cations or outer cations of the metal-hexacianocabaltate (DMC) forms an intermediate oxo-metal (V) in a pre-equilibrium stage, which allows the oxidation of alkene to form the epoxy ring (IV) and regeneration of the DMC Catalyst.

The competence of simultaneous reactions that occur in the reaction to any of the oxidized products with the participation or not of outer cation of the catalyst cannot be ignored. The selectivity towards

the different products is strongly linked to the nature of the outer cation. A comparison of the oxidation performance of the different catalysts (Entries 6–9) revealed the following important information:

"1" Among the catalysts studied, the catalyst containing Cu (Entry 9) as the outer cation in the DMC showed the best yield, the highest conversion (95%), selectivity toward styrene oxide of 64%, and toward benzaldehyde of 32%. The catalyst that contained Ni as an outer cation (Entry 8) also showed higher efficiency in the conversion of styrene, with selectivity toward the same products of 52% and 47%, respectively.

"2" The catalysts with Mn and Co as outer cations (Entry 6–7) showed conversions no higher than 45% of styrene with yields no higher than 12% for SO (IV). Both showed catalytic activity equivalent to the precursor salt ($K_3[Co(CN)_6]$) (Entry 11). This suggests the non-intervention (or too little) of this outer cation during the epoxidation process of the olefin.

Of course, the selectivity towards Bzh (II) is one of the highest, with values close to 75% for both cations. This is explained by the olefin oxidation process in the presence of TBHP, without the participation of a catalyst, Scheme 4.

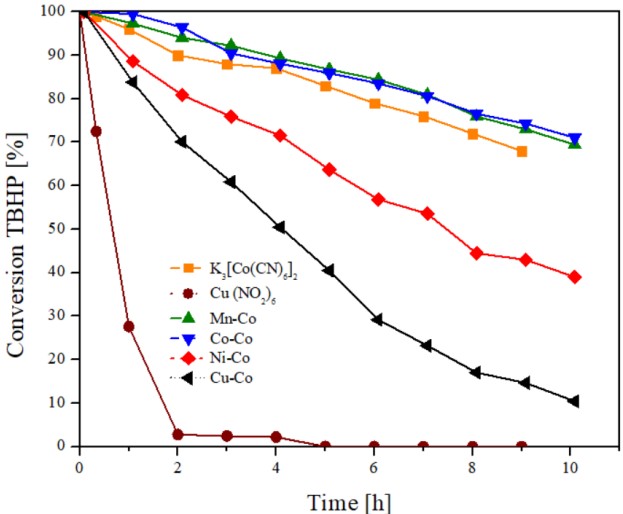

**Scheme 4.** Proposed mechanism of oxidation of olefin without participation a catalyst.

"3" The precursor salt $Cu(NO_2)_6$ like a catalyst (Entry 11) showed a conversion of 68% in only four hours but with the total consumption of the oxidizing agent (TBHP). However, the selectivity and conversion towards the IV was much lower than its homologue Cu in DMC (Cu-Co), Entry 9. Likewise, the salt is irreversibly oxidized by TBHP, incapacitating it to promote the catalytic cycle and consequently the direct oxidation of olefin to Bzh with the subsequent oxidation to carboxylic acid becomes important.

### 2.3. Impact of Outer Cation in DMC in Presence of TBHP

The impact of the outer cations presented in the family of catalysts including the presence of TBHP was studied. Figure 8 shows the disappearance of the oxidizing agent (TBHP) against the different catalyst in function of the time, during the conversion of styrene to oxidized products. The graph shows the conversions corresponding to entries 6–11 of Table 3.

**Figure 8.** Oxidation effect. Exhaustion of the oxidizing agent with styrene TBHP 75 °C, in presence of catalyst.

Further, it can be seen (Figure 8) that a decrease in the rate of decomposition of TBHP occurs as the electronegativity of the outer cation is decreased. The curves that present minor slop, correspond to the outer cations Mn and Co and even the precursor salt of the hexacian block ($K_3[Co(CN)_6]$). In these, the oxidation reaction favors the direct formation of benzaldehyde, due to the poor formation of the intermediate (V) as there are no interaction with the oxygen atom of TBHP.

The curves with a greater slope favor the formation of epoxide (IV), which follows the order of electronegativity of the outer cation; Ni. (1,502) and Cu (1,517).

The rapid consumption of TBHP (2h) in the reaction that contains the simple salt $Cu(NO_2)_4$, shows the low catalytic control due to the deactivation of the Cu when it is rapidly oxidized, this is observed visually during the reaction, with the evident change in coloration from blue to green of the copper in the solution.

At this we could ask, what is the effect that the outer cation has on the DMC to privilege any of the oxidized products?

If the reaction conditions are equivalent in all systems, considering the same oxidation state, we can assume that is the electronegativity together with polarizing power of the outer cation that influences the observed results considerably.

Figure 8 helps to understand the power of these forces, it is possible to observe that the depletion of TBHP is relative to the catalytic efficiency and selectivity of oxidized products. The possible explanation for this is that the oxygen atom, coming from the oxidizing agent (TBHP) binds to the catalyst, therefore promoting an intermediate species that depends on the electronegativity force of each cation. Thus, the more electronegative and more polarizing cations more easily form the intermediate species capable of intervening the catalytic cycle to favor the formation of the epoxide (see Scheme 3).

The behavior of such intermediary specie (V) is not different from the stability conferred by the most electronegative cation according to the Irving-William order [22]. According to which, the stability of complexes increases in the order:

$$Mn^{2+} < Co^{2+} < Ni^{2+} < Cu^{2+}$$

Even when the DMC is formed, it maintains the same order of electronegativity and the outer cations are able to momentarily attract the nucleophilic oxygen atom from TBHP. In this way, the less electronegative cations (Mn and Co), have a weak attraction to the oxygen of TBHP and consequently the formation of the intermediate specie (V) is limited, favoring the direct formation of Bzh (see Scheme 4.

### 2.4. Kinetic Analysis and Effect of the Outer Cation in Catalysis

Kinetic analysis was carried out for all the hexacyanocobaltates as catalysts of the oxidation of styrene and is valid only for reactions in which the main product is oxirane (Entries 6–9). The outer or internal diffusional effects were ignored in this work. For all catalysts, we supposed that the kinetic equation was equal to $r = k[styrene]^{\alpha}[TBHP]^{\beta}[catalyst]^{\delta}$ for the oxidation of styrene, and we considered that the reaction was pseudo first-order, according to what is reported with similar materials [19]. To follow the consumption of styrene, Gas Chromatography-Mass Spectrometry (GC-MS) was used to monitor the reaction kinetics, and the degree of reaction was obtained. To obtain the velocity constant (k) of each of the reactions, we used the expression of concentration of reactants as a function of time for first order reactions, as follows:

$$\ln \frac{[A]}{[A-x]} = kt \tag{1}$$

$$A = [styrene], k_{obs} = k[catalyst]$$

Figure 9 displays the graph obtained from plotting expression (1) versus reaction time (t). The linearity adjustment to the curves for each catalyst shows that it was indeed a pseudo-first-order

oxidation with respect to styrene and TBHP. Table 4 summarizes the fitting parameters and the correlation coefficients, where k (velocity constant) values were the slopes of these fittings.

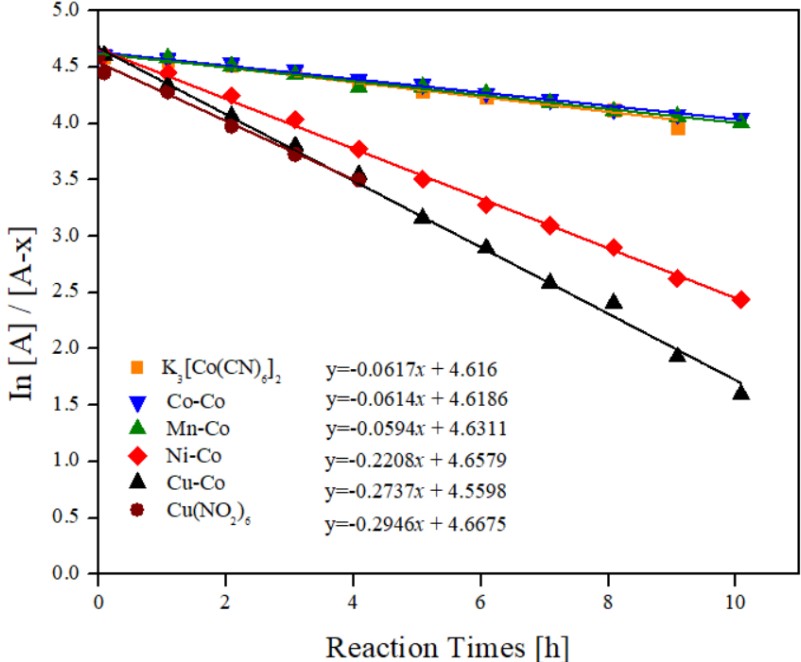

**Figure 9.** Graphs of ln [A]/[A–x] vs. reaction time with Mn–Co, Co–Co, Ni–Co, and Cu–Co catalysts at 75 °C.

**Table 4.** Fitting parameters of the experimental points in Figure 9.

| Entry | Catalyst | Slope (k) | Intercept | R-Square |
|-------|----------|-----------|-----------|----------|
| 1 | Mn–Co | 0.06 | 4.63 | 0.99 |
| 2 | Co–Co | 0.06 | 4.62 | 0.99 |
| 3 | Ni–Co | 0.22 | 4.66 | 0.99 |
| 4 | Cu–Co | 0.27 | 4.67 | 0.99 |
| 5 | $K_3[Co(CN)_6]$ | 0.06 | 4.62 | 0.99 |
| 6 | $Cu(NO_2)_6 \cdot 6H_2O$ | 0.29 | 4.56 | 0.99 |

Reaction conditions included the following: Catalyst, 0.08 mmol; styrene, 1mmol; TBPH, 1.5 mmol; solvent, acetonitrile (5 mL); reaction temperature, 75 °C.

It is possible to observe in the graphs of Figure 9 how the velocity constants vary for each different outer cation? The velocity constants of the catalysts with Mn and Co outer cations (Entry 1 and 2, Table 4) remained very close to each other, which are practically identical to the precursor salt (Entry 5, Table 4). This result indicates no participation in the internal cation in the catalytic activity of these complexes during the epoxidation process. The velocity of reactions catalysed by Ni and Cu (entries 3 and 4) move away considerably, being the catalyst with Cu the one that shows a marked difference in the velocity respect to the other catalysts, approximated 5 times greater than Mn or Co and 2.6 greater than Ni. Thus, epoxidation of styrene by M-hexacyanocobaltates necessarily pass through the formation of an oxo-metal (V) that would be related to the nature of the outer cation.

On the other hand, the superiority in the reaction rate with Cu over its counterparts to oxidize preferentially towards oxirane is remarkable. This can be explained, in addition to the electronegativity already discussed because the Cu has the tendency to capture electrons and thus fill their last valence layer to acquire a configuration of $3d^{10}$. This situation favors, in the case of copper, the speed of the oxidation reaction process. Once the oxo-metal intermediate species is formed (V), allows adding the atom of oxygen to olefin rich in electron density to form the epoxide.

## 3. Experimental

### 3.1. Materials

Potassium hexacyanocobaltate (III) $K_3Co(CN)_6$ ($\geq$ 90%), was procured from Biotechnology Inc. Styrene from Sigma-Aldrich and were used without further purification. Nitrate trihydrate of Cu(II); $Cu(NO_3)_6 \cdot 3H_2O$ ($\geq$ 99.5%), nitrate hexahydrate of Co(II); $Co(NO_3)_6 \cdot 3H_2O$ ($\geq$ 99.5%), nitrate hexahydrate of Ni(II); $Ni(NO_3)_6 \cdot 6H_2O$ ($\geq$ 99.5), sulfate hydrate of Manganese (II); $Mn(SO)_4 \cdot H_2O$ ($\geq$ 99.5%), acetonitrile, tert-butyl hydroperoxide (TBHP), (70%), $H_2O_2$ (50%), were all analytical grate reagents from Sigma-Aldrich.

### 3.2. Catalyst Preparation

$M_3[Co(CN)_6]_2$ catalysts were prepared by co-precipitation method that consists in integrating two solutions by slow dripping, as describe below. First, the solution A was prepared with potassium hexacyanocobaltate as the assembly block in water (100 mL), obtaining a final concentration of 0.01 mmol. On the other hand, the solution B was prepared with copper (II) nitrate tri hydrate, until to achieve an aqueous solution with a concentration of 0.01 mmol. Solution A and solution B were integrated at the same time in a flask, using a peristaltic pump, where a precipitate was immediately formed. After the combination stirring continued for another 1 hour plus. The suspension was maintained for 24 hours without magnetic stir. The resulting product was washed with distilled water and centrifuged several times to remove the accompanying ions. Finally, the solids were vacuum-dried at 50 °C. Under the same way, three more materials were prepared, $Mn_3[Co(CN)_6]_2 \cdot 13H_2O$, $Co_3[Co(CN)_6]_2 \cdot 14H_2O$ and $Ni_3[Co(CN)_6]_2 \cdot 14H_2O$. An overview of the identification of different catalysts is given in table 1 for the M(II)-Co(III) catalysts.

### 3.3. Characterization Apparatus

The X-ray diffraction patterns (XRD) were collected on a Bruker D8 Advance X-ray diffractometer, using the Cu $k_\alpha$ wavelength ($\lambda$ = 1.5406 Å) the Bragg-Bretano geometry was used with a Ge chromator (111) Johansson in the primary beam and a LynxEye detector in the secondary beam. The angular range was from 5° to 90° with a step size of 0.02°.

Scanning electron microscopy (SEM) was carried out on a JEOL 6390 microscope. The same was put over carbon tape and coated with gold before to take the micrographs with 20 kV. Infrared spectra (IR) were recorded in a Spectrum One of Perkin Elmer in the region of 4000–400 $cm^{-1}$ by ATR. TG curves were collected from 25 °C up to 300 °C, under a $N_2$ flow (25 mL/min) using a TA instrument thermo-balance (TGA 2950 model) operated in the high-resolution mode. Gas chromatograph-mass spectrometer (GC-MS, Perkin Elmer Auto system XL Chromatograph/Perkin Elmer Auto Mass under $EI^+$ analysis equipped with a column having a programmed oven (temperature range 323–573 K) were used to identify the mixture of products in the reaction. The conversion and product selectivity were monitored by gas chromatograph (Agilent 6890) equipped with a flame ionization detector, a PE-5 capillary column (30 m-long, 0.32-wide, with a 0.25 μm-thick coated film), a programmed oven (temperature range 323–523 K), He as the carrier gas using bromobenzene as internal standard.

### 3.4. Catalytic Reactions

The hexacyanocobaltates catalyst was used for the catalytic studies without activation. In a typical reaction, 0.08 mmol of catalyst, 1.0 mmol of styrene, 1.5 mmol TBHP (as oxidant) were added to a 200 mL round bottom flask in 5 ml of acetonitrile. The flask was equipped with a cool-water condenser and was maintained in a glycerin bath at 345 K under continuous stirring. The progress of the reaction was monitored by gas chromatograph at intervals. After 10 hours of reaction, the catalyst was separated by centrifugation.

The products are identified by analyzing the reaction mixture with a gas chromatograph-mass spectrometer (GC-MS, Perkin Elmer Auto system XL Chromatograph/Perkin Elmer Auto Mass

spectrometer under EI+ analysis, equipped with a column EN5MS 30m I.D. 0.25 mm film 0.25 μm having a programmed oven (temperature range 323–573K). The conversion and product selectivity were monitored by gas chromatographic analysis with bromobenzene as internal standard. We used a gas chromatograph (Agilent 6890) equipped with a flame ionization detector a PE-5 capillary column (30 m-long, 0.32 mm-wide, with a 0.25 μm-thick coated film), a programmed oven (temperature range 323–523 K), He as the carrier gas.

## 4. Conclusions

A family of hexacyanocobaltates catalysts belonging to the DMC were prepared varying the outer cation. To gain better understanding about their catalytic properties in an epoxidation reaction, the characterization of these materials was made by ATR-FT-IR, TG, EDS, XRD and SEM. It was found that crystallite sizes are strongly depend on the nature of the outer cation. The family of synthesized catalysts maintains a cubic geometry independently of crystallite size. In this case, the crystallite size does not make the catalytic process more efficient because the electronic environment is different in each one of the catalysts evaluated.

In the proposed mechanism based on experimental observations, the oxidation of styrene by DMC catalysts in presence of TBHP as oxidizing agent can be accompanied by two processes: (*i*) The formation of a oxo-metal intermediate, which allows the oxidation of the alkene to form the epoxy ring and, (*ii*) direct oxidation by passing oxirane. These reactions depend on the nature of the outer cation of the catalyst and its concentration in the reaction medium.

Efficiency formation of oxo-metal depends largely of the electronegativity and polarizing power of the outer cation in DMCs, once formed it functions as a donor-acceptor spice, leading to catalysis which can influence the course of the reaction according to factors like electronegativity of the outer cation and concentration in the reaction media.

When the oxo-metal complex is very weak, due to the low electronegativity of the outer cation in the catalyst (Mn, Co) or when its concentration is very low, the reaction in competition, oblivious to oxo-metal, occurs more importantly and determines the process.

**Author Contributions:** P.M.-M. did most of the provision of study materials, designed the catalytic studies and was responsible for writing. R.R.-G., performed design of methodology and was responsible for writing. C.H.-F. helped in the characterization and the writing.

**Funding:** This research received no external funding.

**Acknowledgments:** M.P. acknowledges the Consejo Nacional de Ciencia y Tecnología (CONACYT) and Instituto Politécnico Nacional (IPN) (PIFI Project 20181392 and 20196368) for financial support. The authors would like to Laboratorio Nacional de Conversión y Almacenamiento de Energía (LNCAE).

**Conflicts of Interest:** The authors declare no conflict of interest.

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
