# Peer review of "Role of the Outer Metal of Double Metal Cyanides on the Catalytic Efficiency in Styrene Oxidation"

_catalysts, doi:10.3390/catal9110905_

Round 1

Reviewer 1 Report

In this work, a series of Double metal cyanides catalysts have been synthesized, characterized, and tested their catalytic properties towards the styrene oxidation reaction. Detailed reaction optimization and kinetic studies have been done. However, the work is not presented very well and some major concerns should be addressed:

According to the crystal structure of K3Co(CN)6 (Acta Cryst. (1998). C54, IUC9800062), it is a non-porous structure. Is the K3Co(CN)6 soluble in reaction condition? In any case, the different solubility and porosity makes the comparison between K3Co(CN)6 and other M3[Co (CN)6]2 not fair. This paper also uses wrong formula of “K3[Co(CN)6]2·6H2O” which should be corrected. The mechanism should be reconsidered, from the results, the reactivity is dependent to outer metal, this is understandable since the outer metal is coordinatively unsaturated while the inner Co is octahedrally coordinated. Thus I suspect the outer metal can also catalyze the reaction, rather than tuning the reactivity of inner metal.

Author Response

Please find enclosed our correct manuscript entitled “Role of the outer metal of DMC on the catalytic efficiency in styrene oxidation”. All the authors have participated in these corrections along with and final design of the work

Here I have briefly described the changes made according to the suggestions made by the reviewers, in hopes fulfilling those expectations.

Thank you very much for your kind attention.

The authors

Reviewer 1

1.The precursor salt of all the synthesized catalysts K3Co(CN)6 is effectively soluble under reaction conditions.

The reactions of any catalyst obtained from this one, in fact, cannot be compared as if this salt were a catalyst. However, the tests that were performed with this precursor were simply to rule out the activity of the cation (Co, octahedral), which is the inner cation in all prepared catalysts.

Salt K3Co(CN)6 certainly does not contain water in its structure, so tables 1, 3 and 4 have been corrected in the manuscript. Indeed, the proposed mechanism considers only the outer cation to be responsible for the catalytic activity. In scheme 3 the coordination of the outer cation to the hexacian block is represented with the gray sphere, while the inner cation is represented with red spheres.

Reviewer 2 Report

The paper of P. Molina-Maldonado describing a catalytic performance of double metal cyanides in n styrene oxidation has a certain interest for researches in inorganic chemistry and heterogeneous catalysis. However, before the publication in “Catalysts” the following issues should be addressed.

1.The statement “As for the catalytic oxidation reactions in olefins promoted by hexacianometalates (III) of 66 divalent transition metals, studies are scarce” (P. 2, line 66) should be supported by the appropriate references.

2. The same is for the next phrase “Although it has been reported that there are a probable 67 intermediary species that functions as a transporter of the oxygen atom to the alkene reference”.

3. How the catalyst selectivities were calculated?

4. The reasons for the proposed reaction mechanism (Scheme 3) should be given more accurately.

5. The conclusions are too short and unsuficient. The impact of the nature of catalysts along with their geometrical characteristics should be mentioned. Additionally, the reaction mechanism claimed by authors is unproven.

6. The paper should be formatted carefully. The different fonts are used for paper.

7. P. 13, line 346 – the phrase in Spain language.

8. There are a lot of misprints in the paper.

9. English should be improved. There are some mistakes in the manuscript.

Author Response

Please find enclosed our correct manuscript entitled “Role of the outer metal of DMC on the catalytic efficiency in styrene oxidation”. All the authors have participated in these corrections along with and final design of the work

Here I have briefly described the changes made according to the suggestions made by the reviewers, in hopes fulfilling those expectations.

Thank you very much for your kind attention.

The authors

Reviewer 2

1 and 2. Lines 66 to 69 of paragraph 2 have been referenced.

3. The selectivities were calculated from the substrate that reacted. The values reported in table 3 have been adjusted for an exact 100%. 4. The argument of the mechanism proposed in schema 3 is further developed.

5. The conclusions have been extended to include the physical characteristics of the materials, so the conclusions are more complete.It is clarified in conclusions that the proposed mechanism is based on experimental observations.

6. The characters have been verified for homogenization throughout the text. 7. The Spanish sentence written by mistake has been deleted. 8. A detailed revision of the manuscript has been made. 9. The manuscript has been revised by appropriate English language service.

please see the new manuscript attached

Reviewer 3 Report

Authors reported double metal cyanides as catalysts for styrene oxidation and pointed out the impact of our cations on high conversion and selectivity. Their investigations on catalytic performance look interesting, but most of catalysts' characterization data haven't been analyzed completely yet in this manuscript, including IR, EDS, XRD, TGA and SEM images. Comments and questions are the following.

1. Catalytic Reactions

Is there any reason for 0.08% instead of 0.1% of catalyst amount?

2. Figure 2

They need any reference for the assignments of cyanide band frequency. What are the sharp peaks at around 1650 cm-1 for? In addition, the IR spectrum of K3[Co(CN)6] 6H2O mentioned in Table 1 is missing. 

3. Figure 4

Only spectra are not enough for supporting the ratio of M:Co. They need to include mini quanta with atomic percentages on the spectra. It is quite easy to add them on the program they used for EDS measurements.

4. Figure 5

They just mentioned TGA curves are in Figure 5 without any further interpretation or evaluation.  

5. Figure 6

XRD spectra were obtained, but their peaks haven't been analyzed in details such as peak positions or assignments. Simply mentioning their databases isn't enough.

6. Figure 7

They need to provide the particle size distributions from SEM images for the comparison with the crystallite sizes calculated fro their XRD data in Table 2. Also it is required to mention which XRD peaks were used for the Scherrer equation. 

7. Figure 8 

What is the unit for Y axis, conversion? Is it %?

8. Figure 9

It is strongly recommended to match the colors and symbols in both Figures 8 and 9 for the same legends.

9. Typos

Page 12, line 309: ... like a catalyst (entry 12).. to entry 11

Page 13, line 346: what is the Spanish sentence for?

Author Response

Please find enclosed our correct manuscript entitled “Role of the outer metal of DMC on the catalytic efficiency in styrene oxidation”. All the authors have participated in these corrections along with and final design of the work

Here I have briefly described the changes made according to the suggestions made by the reviewers, in hopes fulfilling those expectations.

Thank you very much for your kind attention.

The authors

Reviewer 3

1.Under general reaction conditions, epoxidations are carried out in the presence of catalysts in concentrations of 1 to 10%, with respect to the initial substrate. 0.08 mmol of catalyst per 1 mmol of styrene was found as an optimized value towards OS.Inconsistencies in the manuscript between mmol and percentages were detected and have been corrected.

2. Reference is provided for allocation bands of the cyanide group. The sharp peaks at around 1650 cm-1 have been integrated in the table 1 and the IR spectrum of the precursor salt K3[Co(CN)6] has been included in Figure 2.

3. For supporting the ratio M:Co, the mini quanta with atomic percentages has been added. Calculations have been made with the EDS program as recommended.

4. An appropriate discussion of Figure 5 (TGA) about the stability of the synthesized catalysts has been added.

5. XRD spectra have been analyzed and position of the peaks has been assigned in Figure 6. Particle size distribution from SEM images has been integrated in the figure 7. In the table 2 have been mentioned the peaks employees for the Scherrer equation.

The crystallite values reported in the manuscript were adjusted according to the new calculation obtained with more than one peak.

7. Symbol % has been added to the Y axis of Figure 8.

8. The colors and symbols of figures 8 and 9 have been homogenized for the same legends.

9. The entry corresponding to the catalyst Cu(NO2)6 has been corrected. The Spanish sentence written by mistake has been deleted.

please see the new manuscript attached

Round 2

Reviewer 1 Report

The wrong formula K3[Co(CN)6]2 is still being used, should change it to K3Co(CN)6

Author Response

1. The error in the formula has been corrected.

Reviewer 3 Report

Authors replied the comments and questions and revised their manuscript well according to them. However the IR analysis has to be done again because of data misreading. In addition there are many typos in tables. It is strongly recommended to check all the values in the tables again to avoid any further issue. Comments and questions are the following.

1. Acronym, DMC, isn't recommended for the title. 

2. Most of the IR values in Figure 2 and Table 1 are misread. For example, the peak labeled for 1622 cm-1 of Figure 2D is placed at higher than the peak labeled for 1642 cm-1 of Figure 2E. It doesn't make sense. Another is the peak labeled for 2184 cm-1 of Figure 2D is higher than 2187 cm-1 of Figure 2E. They have to revise the paragraphs in the main text also according to them. 

3. In Figure 2A, there are two more peaks at around 1630 cm-1, but they didn't count for OH mode in Table 1. 

4. The peak at 416 cm-1 isn't clear for Figure 2A, so they need to expand the X axis to 400 cm-1.

5. What is the peak at ~2.15 keV in the EDS spectra of Figure 4?

6. In Figure 7, the inserts for particle size distributions are too small to read any number. Either better resolutions or bigger figures are required.

7. Table 4 also has discrepancy with Figure 9. Slope 0.29 for Cu-Co doesn't match to 0.2737 in Figure 9. They may have more typos in Table 3. Further checking is required.

Author Response

Please find enclosed our correct manuscript entitled “Role of the outer metal of double metal cyanide on the catalytic efficiency in styrene oxidation”. For the roud 2

Here I have briefly described the changes made according to the suggestions made by the reviewer, in hopes fulfilling those expectations.

The acronym in the title has been replaced by extensive description as suggested Figure 2 maintains the actual IR values read from the original spectra. In the treatment by Origin there was error in the arrangement of spectra. This has already been corrected, so the correspondent text maintains its description. The IR spectrum of salt K3[Co(CN)6] corresponding to Figure 2A was re-performed with the moisture-free sample and thus avoid unjustifiable bands. The IR spectrum reported in Figure 2 is presented in the range of 4000 to 400 cm-1. For a better appreciation of the bands near 450cm-1, a window of »500 to 400 cm-1 has been added The peak at 2.15 keV within the EDS images of Figure 4 belongs to the Au cation, its presence is due to the previous spray used to the sample measurement. The clarification is included in the description of Figure 4. The particle size distribution images have been improved as suggested The investment error in the outstanding values of table 4 according to figure 9 has been corrected as indicated.

Note: the references indicated have been completed